# Robust Face Recognition With Adaptive Mining and Margining of Noises and Hard Samples

## ABSTRACT

While margin-based deep face recognition models, such as ArcFace and AdaFace, have achieved remarkable successes over the recent years, they may suffer from degraded performances when encountering training sets corrupted with noises. This is often inevitable when massively large scale datasets need to be dealt with, yet it remains difficult to construct clean enough face datasets under these circumstances. In this paper, we propose a robust deep face recognition model, RobustFace, by combining the advantages of margin-based learning models with the strength of mining-based approaches to effectively mitigate the impact of noises during trainings. Specifically, we introduce a noise-adaptive mining strategy to dynamically adjust the emphasis balance between hard and noise samples by monitoring the model's recognition performances at the batch level to provide optimization-oriented feedback, enabling direct training on noisy datasets without the requirement of pre-training. Extensive experiments validate that our proposed RobustFace achieves competitive performances in comparison with the existing SoTA models when trained with clean datasets. When trained with both real-world and synthetic noisy datasets, RobustFace significantly outperforms the existing models, especially when the synthetic noisy datasets are corrupted with both close-set and open-set noises. While the existing baseline models suffer from an average performance drop of around 40%, under these circumstances, our proposed still delivers accuracy rates of more than 90%.

## CCS CONCEPTS

• **Computing methodologies → Computer vision problems**.

## KEYWORDS

Face Recognition, Noise Label, Hard Sample Mining, Noise-resistant Loss

## 1 INTRODUCTION

While deep convolutional neural networks (DCNNs) have achieved enormous success in face recognition, its application to massively large-scale datasets still remains a challenging task and hence spaces exist for further research and investigations. The work reported in[29], for example, has indicated that a million-scale face recognition (FR) dataset typically exhibits a noise rate exceeding

*ACM MM, 2024, Melbourne, Australia*

© 2024 Copyright held by the owner/author(s). Publication rights licensed to ACM.
ACM ISBN 978-x-xxxx-xxxx-x/YY/MM
https://doi.org/10.1145/nnnnnnn.nnnnnnn

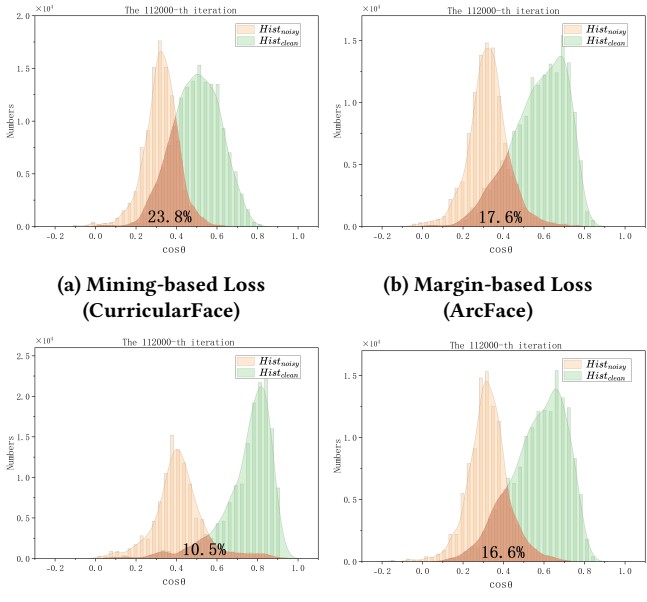

**(a) Mining-based Loss (CurricularFace)**

**(b) Margin-based Loss (ArcFace)**

**(c) Mining-based Antinoise Loss (Ours)**

**(d) Mining-based Antinoise Loss (BoundaryFace)**

**Figure 1: Comparative illustration of the histogram distributions of $\cos\theta$ for different deep models on a training set that is corrupted with 40% close-set noise, where $\theta$ is an angle between the current sample and the class center. As seen, during the initial stages of training, $Hist_{clean}$ and $Hist_{noisy}$ largely overlaps since the models are not trained yet. As training progresses, $Hist_{clean}$ starts to shift to the right, resulting in a reduction of the overlapping region with $Hist_{noisy}$, indicating that the smaller the overlapped region, the less the learning model is affected by noises (*i.e.* more robust to noises). Compared with the other 3 existing models, as seen in fig. 1c, our proposed demonstrates the smallest overlapping region and its corresponding value of $\cos\theta$ closest to 1 on its horizontal axis, indicating that the proposed achieves the highest level of robustness to noises.**

30%, adversely affecting the performance of CNN models. The rapid increase of training datasets, such as Glint360K [2], comprising 360K identities and 18 million images, inevitably introduces noises, making robust face recognition an important topic for intensive investigations and research. Yet elimination of erroneous labels from large-scale FR datasets are both costly and challenging. Despite some efforts proposing automated or semi-automated methods for cleaning large-scale FR datasets [3, 10, 22, 34, 40], the problem still remains non-resolved, indicating that acquisition of large-scale clean datasets with high-quality will involve significant costs and

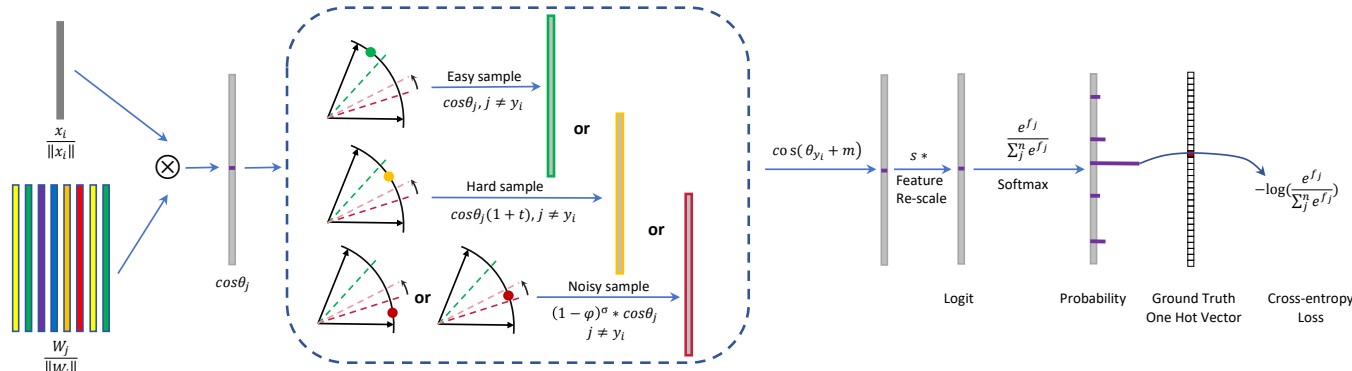

**Figure 2: Illustration of the training process for face recognition by our proposed model RobustFace. Via $l_2$ normalization on both the embedding feature $x_i \in \mathbb{R}^{512}$ and the centers' weight $W \in \mathbb{R}^{512 \times N}$, we get $\cos\theta_j$ (logit) for each class as $W_j^T x_i$. Afterwards, based on the distribution of the angle $\cos\theta_{y_i}$, we classify the samples into easy, hard, and noise samples. By reinforcing the learning on hard samples and weakening the learning on noise samples through the application of applying a modulation function to the negative cosine ($\cos\theta_j, j \neq y_i$), we add an angular margin penalty $m$ to formulate $\cos(\theta_{y_i} + m)$ and multiply $\cos\theta_j$ with the feature scale $s$, and then get the logits to go through the softmax function to construct the cross entropy loss.**

manual annotation is unavoidable. Although learning from noisy datasets has been extensively studied in the field of image classification [8, 9, 11, 24, 26, 28], none of them, however, is applicable to the tasks of face recognition [12]. This is due to the fact that FR datasets often consist of large-scale classes, yet each of which contains relatively few images, making it extremely difficult to identify the relationship patterns from those noisy samples. Consequently, methods designed for classifications cannot be directly applied to FR tasks.

For those massively large scale datasets with the risk of being corrupted by noises, loss design becomes a crucial aspect for break through solutions. Over recent years, state of the arts in loss design is represented by margin based approaches [1, 4, 18, 30], such as ArcFace [6] and AdaFace [15], and mining based approaches [16, 17, 27, 36], such as CurricularFace [14] and BoundaryFace [33], as well as their combined approaches [14, 31]. While margin-based loss aims to enhance the discriminative power by adaptively introducing margins in accordance with the quality of concerned images, all margin-based approaches share the common principle that the importance of hard samples and the impact of noisy samples are relatively overlooked. As a result, recent research [14, 31, 39] has focused on further improving the FR performances through hard sample mining. MV-Arc-Softmax [31] explicitly defines misclassified samples as hard samples and emphasizes their roles by increasing the weight of their negative cosine similarity with a preset constant. In contrast, CurricularFace [14] focuses on curriculum learning, emphasizing those easy samples at early stages and, at later stages, attentions are gradually shifted to those difficult samples. The weakness for mining based loss design, however, lies in the fact that the influence of noise on mining is overlooked. Recent work [5, 7, 9, 12, 33] has begun to design noise-resistant loss functions to mitigate the influence of noises, and one representative example is BoundaryFace [33], which reduces the impact of close-set label noises on training by adding a label correction module. BoundaryFace, however, does not work well for open-set noises in

FR tasks. Sub-center ArcFace [5] introduces sub-centers to relax the intra-class constraint of ArcFace to improve the robustness to label noise. However, while discarding the weak sub-centers containing noisy samples, it also discards some hard samples that have training value. Therefore, the design of a loss function that can perform hard sample mining and tolerate both close-set and open-set noises remains to be an unsolved research problem.

In this paper, we propose a new deep face recognition model: RobustFace, to address the above problem. In comparison with the existing models, it has the feature that training is governed by a noise-adaptive and hard sample mining loss function, which is designed to improve the robustness of deep learning models in the presence of both close-set and open-set noises, enabling direct learning of more effective facial features on large-scale noisy datasets. As shown in fig. 2, this can be verified by observing not only the distribution of the angles between noise and clean samples, but also the distribution of the current class centre at different training stages. Essentially, our proposed utilizes the $\cos\theta$ distribution to "filter out" noise samples and apply a dynamic weighting scheme to enhance the learning process from hard samples while reducing the attention to noise samples. Unlike the existing methods, such as BoundaryFace [33] that requires additional pre-training of the network to handle noises, our proposed RobustFace introduces an adaptive noise handling strategy throughout the entire training process, minimizing the impact of noise from the initial stages on the model. Extensive experiments validate that the proposed RobustFace provides a more stable training under open-set face noises, compared with other existing approaches.

In summary, our contributions can be highlighted as follows:

- We propose a novel antinoise loss function to enable direct and robust training on large-scale noisy datasets, which not only effectively mines cleaner hard samples from noisy training sets, but also reinforce its learning from these samples whilst weakening the learning from noisy samples.

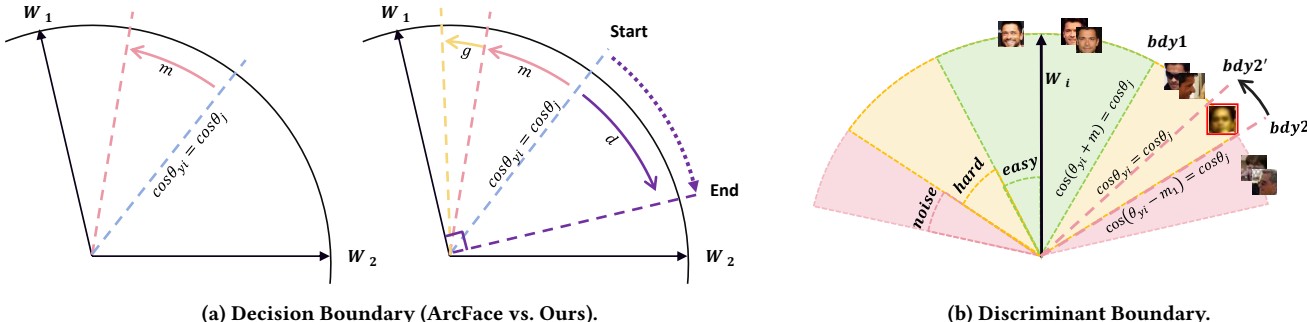

(a) Decision Boundary (ArcFace vs. Ours).

(b) Discriminant Boundary.

Figure 3: (a) illustration of the decision boundaries for ArcFace and the proposed RobustFace, where the blue line, red line and yellow line denote the decision boundary of Softmax, ArcFace and RobustFace, respectively. The purple line is the decision boundary regarding noises for our proposed RobustFace. (b) Illustration of adaptive discriminant boundary for the proposed RobustFace, where those low-confidence noise samples that are unable to promptly optimize towards the class centers are classified as noises.

- We propose a novel paradigm for robust face recognition via monitoring the deep model's recognition ability at the batch level and estimating the noise ratio in the dataset. This proposed paradigm features in serving as a surrogate, where its computational overhead is kept at the minimum.
- Compared with the existing models, our proposed Robust-Face achieves the advantage that, without pre-training, the model is able to filter noises and hence minimizes their impact even from initial stages.
- We have carried out extensive experiments to evaluate the proposed RobustFace, for which the results show that the proposed outperforms the representative existing state of the arts on datasets with closed-set and open-set noises, whilst maintaining competitive performances over those high quality clean datasets.

## 2 THE PROPOSED APPROACH

### 2.1 Adaptive Mining and Margining Loss

Due to the fact that lack of discriminative capability in the learned features prevents the existing softmax loss from achieving effective face recognition in practical applications, a number of variants have been reported, which can be broadly classified into two types: margin-based loss functions and mining-based loss functions. For the convenience of investigations, we summarize all those variants into the following general form:

$$L = -I(p(x_i)) \log \frac{e^{sT(\cos \theta_{y_i})}}{e^{sT(\cos \theta_{y_i})} + \sum_{j=1, j \neq y_i}^{n} e^{sG(t, \cos \theta_j)}} \quad (1)$$

where $p(x_i) = \frac{e^{sT(\cos \theta_{y_i})}}{e^{sT(\cos \theta_{y_i})} + \sum_{j=1, j \neq y_i}^{n} e^{sG(t, \cos \theta_j)}}$ is the predicted ground truth probability, and $I(p(x_i))$ is an indicator function. $T(\cos \theta_{y_i})$ and $G(t, \cos \theta_j)$ are functions that modulate the positive cosine similarity and negative cosine similarity, respectively.

For margin-based loss functions, such as ArcFace, the functions are defined as: $I(p(x_i)) = 1$, $T(\cos \theta_{y_i}) = \cos(\theta_{y_i} + m)$, and

$G(t, \cos \theta_j) = \cos \theta_j$. These functions only modify the positive cosine similarity of each sample to achieve intra-class compactness, whilst the modulation coefficients of each sample's negative cosine similarities are fixed as 1.

For mining-based softmax, such as MV-Arc-Softmax, Curricular-Face, mining essentially emphasizes the value of hard samples, focusing on the relationship between the negative cosine similarity and the positive cosine similarity. That is $I(p(x_i)) = 1$, and thus MV-Arc-Softmax can be formulated as follows:

$$G(t, \cos \theta_j) = \begin{cases} \cos \theta_j, & T(\cos \theta_{y_i}) - \cos \theta_j \geq 0 \\ \cos \theta_j + t, & T(\cos \theta_{y_i}) - \cos \theta_j < 0 \end{cases} \quad (2)$$

and CurricularFace formula is defined as follows:

$$G(t, \cos \theta_j) = \begin{cases} \cos \theta_j, & T(\cos \theta_{y_i}) - \cos \theta_j \geq 0 \\ \cos \theta_j(t + \cos \theta_j), & T(\cos \theta_{y_i}) - \cos \theta_j < 0 \end{cases} \quad (3)$$

As can be seen from the formula above, if a sample is easy, its negative cosine similarity remains unchanged; otherwise, it will be amplified. Specifically, when $T(\cos \theta_{y_i}) - \cos \theta_j < 0$, MV-Arc-Softmax and CurricularFace enhance training on hard samples by amplifying negative cosine similarities with $G(t, \cos \theta_j)$.

To facilitate both intra-class compactness and inter-class separability, we propose a new loss function by adopting an additive angular margin to implement the cosine similarity modulating function $T(\cos \theta_{y_i})$ and maximize the class discriminative ability. In addition, we further introduce a negative cosine similarity modulating function $G(t, \cos \theta_j)$ to mine and reinforce the learning of hard samples while alleviating the adverse effects of noise samples on training. In this way, not only the discriminative capability is enhanced, but also the robustness to noise in the dataset is improved. As a result, the loss function for our proposed RobustFace follows the general form as described in eq. (1), in which $I(p(x_i)) = 1$, and

details of the modulating functions are given below:

$$T(\cos\theta_{y_i}) = \cos(\theta_{y_i} + m)$$

$$G(t, \cos\theta_j) = \begin{cases} \cos\theta_j, & f_{easy} \\ \cos\theta_j(1+t), & f_{hard} \\ (1-\varphi)^\sigma \cos\theta_j, & f_{noise} \end{cases} \quad (4)$$

where $\varphi$ represents a dynamic parameter, which is introduced to provide an adaptive control over the influence of noise in accordance with the training progress. Details of its determination are provided in section 2.2. as shown in eqs. (5) and (6), and $\sigma$ is the modulation parameter set to the default value of 2.

As mentioned in the preceding equations, our proposed model exhibits a flexible decision boundary to address diverse scenarios. In order to maximize the class discriminative capability, as seen in fig. 3a, ArcFace introduces a margin function $T(cos\theta_{y_i}) = cos(\theta_{y_i} + m)$ from the perspective of positive cosine similarity. This is illustrated by the transition from the blue line to the red line in fig. 3a, where $m$ denotes the angular margin added by ArcFace. To enhance its capability in anti-noise, we further introduce an additional margin, denoted as $g = \cos\theta_j(1+t)$, from the negative cosine similarity for hard samples. Correspondingly, the decision boundary is defined as $cos(\theta_{y_i} + m) = cos\theta_j + tcos\theta_j$ (see the yellow line in fig. 3a), reinforcing the learning of hard samples and further improving class discriminability. In addition, we adaptively adjust the weights of noise samples during different training stages. This adaptation leads to the evolution of the decision boundary $cos(\theta_{y_i} + m) = (1 - \varphi)^\sigma \cos\theta_j$ (the blue line in fig. 3a), which gradually reduces the emphasis on noise samples as the model's recognition capability improves. Consequently, the decision boundary experiences the transition towards the purple line, forming a dynamic margin defined as $d = (1 - \varphi)^\sigma \cos\theta_j$, details of which are illustrated in fig. 3a.

## 2.2 Improved Discrimination between Hard and Noise Samples

**Dynamic control over the training progress.** To enable a dynamic adjustment of the noise discriminant boundary and the importance weighting over samples, we introduce a parameter $\varphi$ in eq. (4) to monitor the training progress. By continuously monitoring the training indicator at the batch level, the proposed RobustFace can dynamically adapt its training strategies in response to those ongoing changes and performances, and hence optimizing its ability to handle noise and hard samples during the training process. Although other existing evaluation metrics could also be considered to formulate the training indicator, there exist, however, some limitations that prevent from their utilization. While loss values, for example, are able to measure the differences between the predicted and the targeted, they cannot intuitively demonstrate the ongoing recognition performances during the process of training. Other metrics, such as AUC-ROC, Recall, and F1-Score etc., have the problem that they can not be monitored at batch level, yet incurring significant computational overheads, especially for face recognitions upon massively large scale datasets. To this end, we propose to use the ratio of cosines, which satisfy the condition: $\cos(\theta_j) \geq \cos(\theta_{y_i} + m)$, to the total cosines $C_{(\cos\theta_j)}$ in each

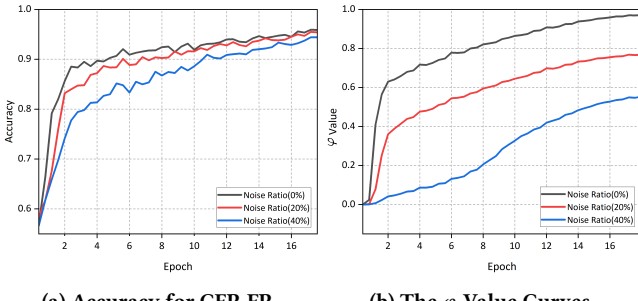

(a) Accuracy for CFP-FP.     (b) The $\varphi$-Value Curves.

Figure 4: (a) Illustration of the varying performances by ResNet-34, which is trained by the proposed RobustFace with different ratios of close-set noises (*i.e.* 0%, 20%, 40%). (b) illustration of how the values of $\varphi$ varies when the proposed RobustFace is trained with different ratios of close-set noises (*i.e.* 0%, 20%, 40%)

batch as the training indicator. In this way, the recognition performances at the batch level can be intuitively monitored during the training process, yet the incurred computational overhead remains negligible, details of which are given below:

$$\varphi = \frac{1}{n}\sum_{i=1}^{n}\widehat{\varphi_i}, \quad \widehat{\varphi} = \frac{C_{(\cos\theta_j > \cos(\theta_{y_i}+m))}}{C_{(\cos\theta_j)}}(1 - \mu) \quad (5)$$

where $n$ is the batch size, $\mu$ represents the prior noise rate of the current training set. As $\mu$ normally has minor impacts on the model's results under low noise conditions, it is commonly set to $\mu = 0$, that is, $\widehat{\varphi} = \frac{C_{(\cos\theta_j > \cos(\theta_{y_i}+m))}}{C_{(\cos\theta_j)}}$.

When the batch size is small, the batch statistics $\varphi$ might be unstable. To address this, we employ exponential moving average (EMA) across multiple batches to stabilize $\varphi$. Specifically, let $\varphi^{(k)}$ denote the $k$-th step batch statistics of the training indicator, we have:

$$\varphi = \alpha\varphi^{(k)} + (1 - \alpha)\varphi^{(k-1)} \quad (6)$$

where $\varphi$ is adjusted by using a momentum $\alpha$, which is set to 0.99.

To indicate the ongoing status of training progress, $\varphi$ is required to assess model performances, enabling efficient monitoring at the batch level with minimal computational overhead. From fig. 4, it can be observed that the curves of $\varphi$ (fig. 4b) exhibits a shape and trend similar to that of the accuracy curve on the validation set (fig. 4a) when the training set contains varying degrees of noises ( 0%, 20%, and 40% closed-set noise). These findings provide compelling evidences that $\varphi$ is indeed an effective indicator in reflection of the ongoing status of the model's training progress. In addition, it is also found that the convergence upper limit of $\varphi$ reflects the noise rate of the current training set as shown in fig. 4b. Hence, this approach can be used to approximate the value of the prior noise rate $\mu$. While the model's final performance is typically not sensitive to the value of $\mu$, a more accurate estimation of $\mu$ can stabilize the training process, and thus improve the performances, especially in the presence of significant noises. Therefore, if one desires to train the model using a precise prior noise rate $\mu$, the following steps can be taken: 1) Set $\mu$ to the default value of 0 and

**Table 1: Experimental results in recognition accuracy rates upon the varying values of $m_1$ in close-set noisy environments.**

| Method(%) | Noise Ratio | AgeDB [21] | CALFW [38] |
|---|---|---|---|
| $m_1$=0.10 | | 94.35 | 93.96 |
| $m_1$=0.15 | 0% | **94.65** | **94.11** |
| $m_1$=0.20 | | 94.47 | 93.90 |
| $m_1$=0.10 | | 93.53 | 93.45 |
| $m_1$=0.15 | 20% | 93.72 | **93.49** |
| $m_1$=0.20 | | **93.79** | 93.47 |
| $m_1$=0.10 | | 91.84 | 92.25 |
| $m_1$=0.15 | 40% | **92.22** | **92.47** |
| $m_1$=0.20 | | 92.17 | 92.33 |

**Table 2: Experimental results in recognition accuracy rates upon the varying values of $t$.**

| Method(%) | CFP-FP | CALFW | CPLFW |
|---|---|---|---|
| $t = 0.20$ | **91.81** | 94.47 | **88.11** |
| $t = 0.25$ | 91.78 | **94.50** | **88.11** |
| $t = 0.30$ | 91.31 | 94.44 | 87.86 |

train the model. 2) Estimate a more accurate value of $\mu$ based on the convergence upper limit of $\varphi$. 3) Adjust the value of $\mu$ accordingly and retrain the model.

**Soft Noise Discriminant Boundary.** Given the difficulties in learning noise samples, the relationship between the positive cosine similarity and the negative cosine similarity reflects the type of samples. In other words, as CNNs can quickly memorize simple/clean samples and eventually learn difficult/noisy samples, the noise samples may be distributed along the nearest negative class discriminant boundary (i.e. $\cos \theta_j = \cos \theta_{y_i}$). On this basis, an additional soft buffer margin $m_1$ can be introduced as the soft discriminant boundary between hard and noise samples. Therefore, the noise determination condition $f_{noise}$ can be constructed as follows:

$$f_{noise} : \exists j \in \{1, 2, \ldots, n\} \setminus \{y_i\} : \cos \theta_j \geq \cos(\theta_{y_i} - m_1) \quad (7)$$

Where, $m_1$ is the updated additional soft buffer margin and calculated by: $m_1 = (1-\varphi)^2 \cdot \widehat{m_1}$, and $\widehat{m_1}$ is an original soft buffer margin. This formulation ensures that, as the training indicator $\varphi$ improves, the soft buffer margin $m_1$ gradually decreases, allowing for a more precise discrimination between hard and noise samples with low confidences.

Specifically, as shown in fig. 3b, easy samples with low separability (green area) have a smaller impact on the model's learning, whilst the high-confidence noise samples are easier to filter. Additionally, low confidence noise samples (red box) close to "bdy2" may be mis-classified as hard samples, leading to a stronger negative impact on the model's optimization. Therefore, correct identification of those low-confidence noise samples is crucial for enhancing

the discriminability of features. To address this issue, we introduce a soft margin $m_1$ to aggressively filter out noise samples without concerning about potential misclassifications caused by the model's early recognition capability limitations. As the model's recognition capability, indicated by $\varphi$, improves, the confidence in noise determination also increases. As a result, the "noise" decision boundary gradually transitions from $\cos(\theta_{y_i} - m_1) = \cos(\theta_j)$ to $\cos(\theta_{y_i}) = \cos(\theta_j)$, thereby reassigning these samples as "noise" and reducing the adverse effects caused by mis-classifications. By employing this method, the proposed RobustFace is able to focus on learnings from informative features (i.e., hard samples), reducing its reliance on harmful features (i.e., noise samples), and accelerating the model's learning of correct features or patterns throughout the entire training phase.

Given the fact that easy samples are closer to the class center compared to hard samples, the condition $\cos \theta_j \leq \cos(\theta_{y_i} + m)$ will hold even with an added penalty margin $m = 0.5$. Therefore, the noise determination condition $f_{easy}$ can be formulated as:

$$f_{easy} : \exists j \in \{1, 2, \ldots, n\} \setminus \{y_i\} : \cos \theta_j \leq \cos(\theta_{y_i} + m) \quad (8)$$

Similarly, the noise determination condition $f_{hard}$ can be formulated as:

$$\begin{aligned} f_{hard} : &\exists j \in \{1, 2, \ldots, n\} \setminus \{y_i\} : \\ &\cos(\theta_{y_i} - m_1) > \cos \theta_j > \cos(\theta_{y_i} + m) \end{aligned} \quad (9)$$

## 3 EXPERIMENTS

### 3.1 Implementation Details

**Datasets.** Refined from the MS-Celeb1M[10], MS1MV2 [6] is a widely used clean dataset for face recognition training, which contains 85K identities and over 5.8M images. To simulate the datasets with various levels of noises, synthetic datasets are created based on MS1MV2. For close-set noises, the labels of MS1MV2 samples are randomly flipped, and for open-set noises, Glint360K [2] is selected as the source, and MS1MV2 samples are randomly replaced. In contrast, the original MS-Celeb-1M dataset is a large-scale dataset of facial images, encompassing various poses, expressions, and complex lighting conditions (100k identities, 10 million images). Unlike the refined MS1MV2, MS-Celeb-1M is untreated, which consists of many noisy faces, serving as the training set to evaluate performances under realistic noise distributions. Based on the prior noise estimation described in section 2.2, we estimate that the noise rate in MS-Celeb-1M is approximately 40%, which closely aligns with the findings in [29]. In summary, we evaluate the proposed RobustFace on several popular benchmarking datasets, including LFW [13], CFP-FP [25], CPLFW [37], AgeDB [21], CALFW [38], IJB-B [32], and IJB-C [19].

**Training Setup.** To pre-process the datasets, we follow the work [35] to align and crop the 112×112 faces with five landmarks, and employ ResNet with a 512-dimensional feature output as the backbone. While the framework is implemented using PyTorch [23], the model is trained on 8 NVIDIA A100-PCIE GPUs with a batch size of 128*8, and the training is finished at 20 epochs. By setting the scale at $s = 64$, trainings are performed using the SGD algorithm with a momentum of 0.9 and weight decay of 5$e$-4. Our source codes will be made available after accepted.

**Table 3: Experimental results in recognition accuracy rates achieved by all assessed models when trained on real-world clean and noisy datasets. [\*=our evaluation of the released model]**

| Method(%) | Train Data | LFW [13] | CALFW [38] | AgeDB [21] | CFP-FP [25] | CPLFW [37] | AVG |
|---|---|---|---|---|---|---|---|
| ArcFace [6] | MS1MV2 | 99.82 | 95.45 | 98.28 | 98.27 | 92.08 | 96.78 |
| MV-Softmax [31] | MS1MV2 | 99.80 | 96.10 | 97.95 | 98.28 | 92.83 | 96.99 |
| CurricularFace [14] | MS1MV2 | 99.80 | 96.20 | **98.32** | 98.37 | 93.13 | 97.16 |
| Sub-center Arcface*[5] | MS1MV2 | 99.81 | 95.63 | 98.19 | 98.31 | 92.58 | 96.90 |
| MagFace [20] | MS1MV2 | **99.83** | 96.15 | 98.17 | 98.46 | 92.87 | 97.10 |
| AdaFace [15] | MS1MV2 | 99.82 | 96.08 | 98.05 | **98.49** | **93.53** | **97.19** |
| BoundaryFace*[33] | MS1MV2 | 99.81 | 96.14 | 98.17 | **98.49** | 93.14 | 97.15 |
| **RobustFace**,R100 | MS1MV2 | 99.82 | **96.23** | 98.19 | 98.39 | 93.03 | 97.13 |
| ArcFace*,R50 | MS-Celeb-1M | 99.66 | 97.30 | 96.82 | 95.37 | 91.27 | 96.08 |
| CurricularFace*,R50 | MS-Celeb-1M | 99.74 | 97.54 | 96.82 | 95.58 | 92.39 | 96.41 |
| Sub-center Arcface*,R50 | MS-Celeb-1M | 99.71 | 98.01 | 96.95 | 95.59 | 92.41 | 96.53 |
| AdaFace*,R50 | MS-Celeb-1M | 99.69 | 97.47 | 96.97 | 95.39 | 92.14 | 96.33 |
| BoundaryFace*,R50 | MS-Celeb-1M | 99.72 | 97.41 | 96.96 | 95.59 | 91.39 | 96.21 |
| **RobustFace**,R50 | MS-Celeb-1M | **99.76** | **98.41** | **97.55** | **95.87** | **93.33** | **96.98** |

**Table 4: 1:1 verification TAR@FAR on the datasets: IJB-B and IJB-C.**

| Method(%) | IJB-B [32] (TAR@FAR) | | IJB-C [19] (TAR@FAR) | |
|---|---|---|---|---|
| | 1e-5 | 1e-4 | 1e-5 | 1e-4 |
| ArcFace | 63.35 | 85.03 | 70.77 | 87.73 |
| CurricularFace | 74.30 | 89.04 | 83.50 | 92.05 |
| Sub-center Arcface | **74.33** | 89.07 | 83.57 | 92.30 |
| AdaFace | 72.83 | 89.03 | 83.55 | 91.95 |
| BoundaryFace | 65.37 | 84.80 | 74.55 | 88.13 |
| **RobustFace** | 73.10 | **91.30** | **85.50** | **94.02** |

## 3.2 Ablation study

**Effect of parameter $m_1$.** As described in the previous section, the soft buffer margin $m_1$ plays a crucial role in the proposed Robust-Face, serving two main purposes: (i) adjusting the boundary for noise sample classification; (ii) dynamically balance the impact of noise during trainings. An appropriate setting of $m_1$ enables the model to focus more on informative features in the presence of noises, and hence enhancing its recognition performances and robustness. Consequently, it becomes important to further explore the optimal value of $m_1$ under close-set noisy environments. As reported in table 1, $m_1 = 0.15$ achieves the best performance at noise rates of 0% and 40%. For the dataset with a noise rate at 20%, $m_1 = 0.2$ demonstrates a slight advantage in AGEDB, whilst $m_1 = 0.15$ remains optimal in CALFW. Via comprehensive consideration of all these settings and their corresponding balances across the varying results, we set $m_1 = 0.15$ in the subsequent experiments.

**Effect of parameter $t$.** As described in eq. (4), the hyper-parameter $t$ is a modulation coefficient of the negative cosine similarity for hard samples, which is introduced to determine an appropriate emphasis on hard samples during the training process. The insensitivity of the proposed RobustFace to the $t$ within a certain range is demonstrated in table 2, based on which we set $t = 0.2$ as the optimal value for RobustFace in subsequent experiments.

## 3.3 Comparative Evaluation Against The Existing SoTA Models

**Learning from MS1MV2, MS-Celeb-1M.** To validate the performance of RobustFace on a clean training set, we carried out extensive experiments in comparison with a number of the existing SoTA models and report their performances on 5 benchmarking datasets described in section 3.1. To provide a comprehensive assessment, we follow the work [6] and select different backbones (ResNet100, ResNet50) to showcase the flexibility and applicability of our proposed method across various backbone architectures. The experimental results are listed in table 3, from which it can be seen that, even for a clean training dataset, the proposed RobustFace achieves competitive performances when compared with those mining-based models, such as CurricularFace [14] and Boundary-Face [33]. As the primary advantage for RobustFace lies in the fact that it is capable of effectively handling noisy training datasets, RobustFace may be disadvantaged in the case of high quality training datasets. When trained with the clean and high quality dataset MS1MV2, nonetheless, RobustFace still achieves the best result on CALFW, the second best result on LFW, and competitive results for all other validation datasets in comparison with the selected 5 existing SoTA models, details of which are indicated by the results listed in the upper part of table 3. When trained with the noisy dataset MS-Celeb-1M [10], the proposed RobustFace achieves the best results across all validation datasets in comparison with ArcFace, CurricularFace, AdaFace, Sub-center Arcface [5] and Boundary-Face, indicated from the results listed in the lower part of table 3. Due to the fact that most of the existing models have reported their experimental results trained on MS1MV2, we are able to assess

**Table 5: Experimental results in recognition accuracy rates achieved by all the assessed models, where the training set is contaminated with different ratios of close-set noises.**

| Method(%) | Noise Ratio | LFW [13] | CALFW [38] | AgeDB [21] | CFP-FP [25] | CPLFW [37] | AVG |
|---|---|---|---|---|---|---|---|
| ArcFace | C-15% | 99.77 | 96.03 | 97.60 | 96.32 | 91.52 | 96.25 |
| CurricularFace | C-15% | 99.73 | 96.08 | 97.68 | 96.11 | 91.09 | 96.14 |
| Sub-center Arcface | C-15% | 99.75 | 96.11 | 97.88 | 96.64 | 91.69 | 96.41 |
| AdaFace | C-15% | 99.67 | 96.07 | 97.81 | 96.75 | 91.71 | 96.40 |
| BoundaryFace | C-15% | **99.78** | 96.00 | 97.93 | **96.78** | 91.62 | 96.42 |
| **RobustFace** ($\mu = 0.15$) | C-15% | **99.78** | **96.13** | **97.99** | 96.75 | **91.73** | **96.48** |
| ArcFace | C-30% | 99.38 | 95.77 | 97.47 | 95.51 | 90.87 | 95.80 |
| CurricularFace | C-30% | 99.04 | 95.76 | 97.04 | 95.43 | 90.87 | 95.63 |
| Sub-center Arcface | C-30% | 99.21 | 95.81 | 97.22 | 95.49 | 90.91 | 95.72 |
| AdaFace | C-30% | 99.59 | 95.77 | 97.27 | 95.43 | 90.60 | 95.73 |
| BoundaryFace | C-30% | 99.23 | 95.83 | 97.23 | 95.37 | 90.43 | 95.62 |
| **RobustFace** ($\mu = 0.30$) | C-30% | **99.61** | **95.91** | **97.51** | **95.69** | **91.09** | **95.96** |
| ArcFace | M-(15%, 5%) | 65.78 | 59.49 | 55.33 | 57.90 | 52.07 | 58.11 |
| CurricularFace | M-(15%, 5%) | 63.14 | 57.75 | 53.90 | 54.32 | 52.32 | 56.29 |
| Sub-center Arcface | M-(15%, 5%) | 73.21 | 69.51 | 59.59 | 64.79 | 61.41 | 65.70 |
| AdaFace | M-(15%, 5%) | 68.83 | 59.49 | 55.82 | 59.27 | 54.78 | 59.64 |
| BoundaryFace | M-(15%, 5%) | 72.01 | 63.21 | 59.69 | 57.20 | 53.54 | 61.13 |
| **RobustFace** ($\mu = 0.20$) | M-(15%, 5%) | **99.71** | **96.04** | **97.54** | **96.58** | **91.43** | **96.26** |
| ArcFace | M-(10%, 10%) | 54.54 | 50.18 | 53.46 | 51.45 | 51.19 | 52.16 |
| CurricularFace | M-(10%, 10%) | 63.43 | 55.86 | 55.18 | 51.91 | 51.26 | 55.53 |
| Sub-center Arcface | M-(10%, 10%) | 69.20 | 67.39 | 57.21 | 60.43 | 56.90 | 62.22 |
| AdaFace | M-(10%, 10%) | 62.70 | 57.76 | 55.38 | 52.13 | 51.71 | 55.94 |
| BoundaryFace | M-(10%, 10%) | 51.53 | 49.98 | 49.41 | 51.75 | 50.06 | 50.55 |
| **RobustFace** ($\mu = 0.20$) | M-(10%, 10%) | **99.75** | **96.08** | **97.62** | **96.14** | **91.28** | **96.17** |

the proposed RobustFace in comparison with all of them without repeating any of their experiments. For the experiments trained on the noisy dataset MS-Celeb-1M, however, none of these existing models have reported any experimental results, and thus making it an enormous task to complete comparative experiments with all of them. To this end, we only selected the best performing models, *i.e.* CurricularFace, AdaFace, and noise-resistant models, *i.e.* Sub-center ArcFace and BoundaryFace, for the subsequent experiments, the results of which are illustrated in the lower part of table 3. Considering the fact that our work started from referencing ArcFace, we also included this model for additional coverage.

Further examination of the results listed in both upper part and lower part of table 3 indicates that, when measured in average recognition rates, our proposed RobustFace only experienced a small drop of 0.15 from being trained on MS1MV2 to that on MS-Celeb-1M, yet in contrast, the drop experienced by the compared benchmarks, say ArcFace, CurricularFace, Sub-center Arcface, AdaFace, and BoundaryFace, can be seen as 0.7, 0.75, 0.37, 0.86 and 0.94, respectively.

To further evaluate the performance of RobustFace on real-world noise datasets, we conducted large-scale 1:1 face verification using ResNet50 trained with RobustFace on the MS-Celeb-1M dataset. For the 1:1 verification task, the number of positive and negative matches is set up as: 10k and 8M in IJB-B, and 19k and 15M in IJB-C, respectively. table 4 presents the experimental results in comparison with the selected existing SoTA models, showcasing their TARs at

different FAR levels, including 1e-5, and 1e-4. As seen, the proposed RobustFace demonstrates a remarkable performance improvement on both IJB-B and IJB-C. When compared with the second best models at FAR=1e-4, for example, the proposed RobustFace achieves approximately 2.23 and 1.72 performance improvement over IJB-B and IJB-C, respectively. Even at FAR of 1e-5, a significantly more challenging level, our proposed still maintains superiority and the improvement achieved is 1.93 over IJB-C.

**Learning from synthetic noise datasets.** To provide a comprehensive evaluation on the proposed RobustFace, we follow the work reported in [33] to apply two types of synthetic noise datasets, close-set noise and open-set noise, to the training of the assessed models. While the close-set noise is introduced by randomly changing the labels of the facial images inside the training dataset, the open-set noise is introduced by changing the label of those facial images that are not included inside the training dataset. In this way, we are able to provide a quantified assessment on the robustness of the compared models and see how they perform under such seriously synthesized noisy environments. To achieve a good balance between computing cost and accuracy, we use the ResNet34. The experimental results are listed in table 5, where the data ratio figures represent the proportion of labels being randomly changed. As an example, C-15% (close-set rate: 15%) means that 15% of facial labels inside the training set are randomly changed, creating the

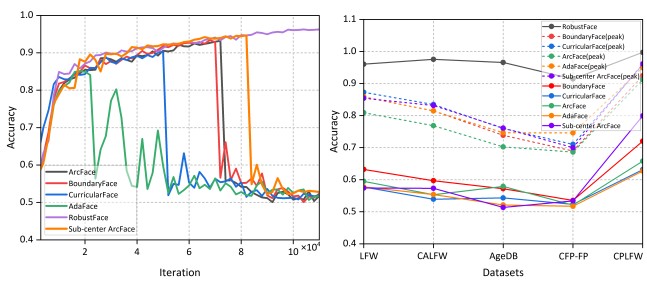

(a) Accuracy for CFP-FP.  (b) Acc-peak vs. Acc-final.

**Figure 5: Illustration of how all the assessed models' performances vary during trainings: (a) the varying results achieved by all models when the training set contains 10% open-set noise and 10% close-set noise. (b) the results in peak accuracy values (Acc-peak) and final accuracy values (Acc-final). While the Acc-peak represents the highest validation accuracy achieved during the training process (dashed line), the Acc-final denotes the final validation accuracy after the loss has stabilized (solid line).**

effect that 15% of the noise being added, since those labels have now been corrupted. As seen in table 5, the proposed RobustFace trained on the datasets with 15% and 30% close-set noise overwhelmingly outperforms the baseline models, indicating its capability to learn more discriminative features from noisy training sets.

Compared with the simple close-set face noise, the open-set face noise is more challenging for face recognition tasks. As seen in table 5, when the training sets are corrupted with 15% close-set noise and 5% open-set noise, *i.e.* M(15%, 5%), all baseline models suffered from significant performance drops, around 40%, across all the five validation datasets, yet in contrast, our proposed RobustFace still maintains robust performances above 90% over all the validation datasets. Even when the corruption percentage is increased to M(10%, 10%), RobustFace still maintans an accuracy of above 90% across all validation sets. Specifically, for the mixed datasets M(15%, 5%) and M(10%, 10%), RobustFace achieves average accuracies of 96.26% and 96.17%, respectively, while the highest accuracies of other methods are only 65.70% and 62.22%.

To find out how the performances of the assessed models emerge and vary over the ongoing training process, we monitored the recognition accuracy rates over every 2000 iterations for all the assessed models, and illustrate the experimental results in fig. 5. As seen from fig. 5a, the baseline models, including ArcFace, CurricularFace, AdaFace, and BoundaryFace, struggle in effectively handling the open-set noise (10%), suffering from significant accuracy drops attributed to overfitting to the noise. Specifically, as the training progresses, the impact of well-separated easy samples on the model diminishes, while misclassified noise samples exert a greater influence. This phenomenon causes the class centers to shift towards wrong directions and gradually move away from those classified samples, resulting in a substantial decline in recognition rates. In fig. 5b, we illustrate the peak accuracy rates (Acc-peak) in dashed lines that are achieved during the training process and the final converged accuracy rates (Acc-final) in solid lines for all

the assessed models across the five validation datasets. As seen, all the existing baseline models exhibit significant gaps between their Acc-peak and Acc-final values. Yet in comparison with the proposed RobustFace as given in the black solid line, the baseline models illustrated significant shortfalls even with their Acc-peak values. As a result, this experiment illustrates that the proposed RobustFace outperforms the baseline models not only in terms of the Acc-final, but also in terms of the Acc-peak ever achieved by those baseline models during the whole training process.

Out of the above experiments, the proposed RobustFace demonstrates the following advantages in comparison with the existing baselines: (i) Capable of highlighting the role of mining those hard samples to enhance the model's discriminative capability, and thus enabling competitive performances even trained on clean datasets (See the results listed in table 3). (ii) Compared with the existing mining-based models, the proposed RobustFace effectively isolates those clean hard samples from noisy training sets to strengthen the learning effectiveness of those samples while adaptively reducing its reliance on noisy samples. Consequently, both the model's discriminative capacity and the robustness to noises are significantly enhanced (See the results given in table 5 and fig. 5). (iii) The proposed RobustFace embeds with an adaptive noise-handling strategy throughout the life cycle of trainings, via which the impact of noises is minimized right from the initial stages of training, and thus a stable training can be maintained even when more complex noises are encountered, such as those mixtures of open-set and close-set noises (See the results given in table 5 and fig. 5).

## 4 CONCLUSIONS

In this paper, we have proposed a robust deep model for face recognitions, where a new loss function is constructed, featuring in adaptive mining of those noise and hard samples. In comparison with the existing approaches, the proposed RobustFace largely alleviates the poor performance of mining-based softmax on datasets with severe noise corruptions, serving as the first known method to effectively address open-set noises. In addition, the proposed RobustFace is easy to implement, and the introduced noise-adaptive mining has negligible computational overhead. Extensive evaluations have been carried out on both real-world and synthetic noisy datasets against the representative existing SoTA models across five validation datasets, which are widely adopted in the field. The experimental results illustrate that our proposed achieved competitive and SoTA performances trained on clean datasets. When trained with noisy datasets, especially with open-set noises, our proposed significantly outperforms all the compared benchmarks by considerable margins.

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
