# OpenReview forum: "RobustFace: Adaptive Mining of Noise and Hard Samples for Robust Face Recognitions"
_acmmm.org/ACMMM/2024/Conference — MM2024 Poster_

### Official Review · Reviewer_VHnV · 2024-05-04

**Rating:** 4
**Confidence:** 3

**Summary:**

This paper focuses on face recognition and proposes RobustFace which combines the advantages of margin-based learning models with the strength of mining-based approaches to effectively mitigate the impact of noises during trainings. It introduces a noise-adaptive mining strategy to dynamically adjust the emphasis balance between hard and noise samples. Experiments on several datasets prove the effectiveness of the proposed method.

**Strengths:**

1. The paper is well written and easy to follow.
2. The fundamental idea of this paper is technically correct.

**Limitations:**

1. In Page 4, Line 455, the authors mentioned that “it is also found that the convergence upper limit of \varphi reflects the noise rate of the current training set.” However, this conclusion has only been demonstrated using the MS-Celeb-1M dataset, as shown in Line 565. It would be better to conduct additional experiments on several synthetic noise datasets with varying noise ratios to validate this claim.
2. How do the hyper-parameters \sigma and \alpha affect the results? How are their values determined? It would be beneficial to include sensitivity analyses of parameters to elucidate this.
3. The authors believe that the model’s final performance is typically not sensitive to the value of \mu; however, a more accurate estimation of \mu can stabilize the training process. Conducting experiments to support this assertion would be advisable.
4. During the training process, is the proposed method accurate in distinguishing between easy, hard, and noisy samples? It is advisable to visualize these results to demonstrate the effectiveness of the method.

**Suitability:**

2

---

### Official Review · Reviewer_ftwD · 2024-05-21

**Rating:** 4
**Confidence:** 4

**Summary:**

This paper introduces a novel robust face loss function designed to address label noise and hard samples in face recognition datasets. The proposed method computes parameters for simple, hard, and noisy samples, assigning weights to the negative cosine values of these samples. This approach effectively emphasizes hard samples and enhances noise robustness. While the experimental results demonstrate the effectiveness of the proposed loss function in datasets containing massive label noise, the performance of training on the low noisy datasets is sub-optimal. Additionally, the paper lacks comparative experiments with some latest label noise algorithms, which could provide further insights into the proposed method's efficacy.

**Strengths:**

1) The paper adopts a novel perspective by utilizing the proportion of misclassified cosine similarities to gauge training progress and subsequently adjusting the weights of negative cosine similarities for negative class samples, thus enhancing the model's robustness to label noise.
2) Through extensive experimentation, the proposed method's outstanding performance in noisy datasets is thoroughly validated.
3) The proposed loss function demonstrates robustness against both closed-set and open-set label noise scenarios.

**Limitations:**

1) The parameters are unstable - The instability of hyperparameters is evident from the results of ablation experiments, particularly in the relationship between the setting of parameter m1 and the noise ratio within the dataset.
2) The expression is unclear- A clarification is needed regarding the statement(Section 2.2): "In addition, it is also found that the convergence upper limit reflects the noise rate of the current training set as shown in fig. 4b." This assertion seems misstated, as the conclusion drawn from the figure appears to pertain to the proportion of clean data rather than the noise rate.
3) The experiment was inadequate - The paper lacks a comparative analysis with recent methods for handling label noise in face recognition, such as RVFace.
4) The symbols are inconsistent：The expression of parameter m1 in the formula 7 is inconsistent with the symbol used in the ablation experiments.

**Suitability:**

3

---

### Official Review · Reviewer_LYTo · 2024-05-21

**Rating:** 4
**Confidence:** 3

**Summary:**

Considering the neglect of label noise in mining-based large-margin loss, this paper proposes RobustFace, with a new loss function that combines margin-based learning and mining-based learning. The proposed method shares a similar formulation of CurricularFace but incorporates an adaptive control over the influence of noise into the modulating function. Experiments are conducted on MS1MV2 and Glint360K for synthetic close-set and open-set label noise respectively, and conducted on MS-Celeb-1M for real-world label noise. Models are mainly evaluated on LFW, CALFW, AgeDB, CFP-FP and CPLFW. IJB-B and IJB-C are also used for the final evaluation.

**Strengths:**

+ The paper is motivated well. The paper pointed out the challenges in mining-based large-margin loss, which only focus on differentiating easy training examples and hard examples, but neglecting the influence of label noise. Considering this point, the proposed RobustFace introduces a new branch of modulating function to address the label noise problem. As the previous works [29] illustrated, label noise robust training is an important topic in face recognition.
+ The paper has conducted a variety of experiments to demonstrate the effectiveness of the proposed method, especially in open-set label noise settings.

**Limitations:**

- Despite the proposed method having been compared with some previous works like Sub-center ArcFace, the paper may neglect some important related work [1] for coping with open-set noise in FR. In addition, it will be better if more detailed experimental analyses are provided. For example, in Figure 1, please consider putting the analysis for the $theta$ of Sub-center ArcFace here.

[*] Dynamic training data dropout for robust deep face recognition, TMM, 2021.

**Suitability:**

3

---

### Official Review · Reviewer_RsDT · 2024-06-05

**Rating:** 3
**Confidence:** 4

**Summary:**

This paper introduces a deep face recognition model, which combines the advantages of margin-based learning models and mining-based approaches, aiming to mitigate the impact of noise during the training process. It also conducts several experiments to validate the effectiveness of the method.

**Strengths:**

This paper aims to address the issue of noisy samples in large-scale face recognition datasets, which has practical significance, and the motivation is quite intuitive.

**Limitations:**

1. The authors have utilized R100, R50, and R34 as CNN backbone networks in Table 3 and Table 5, respectively. It's unclear why there isn't consistency in their choice.
2. The performance of the comparison methods in Table 5 is notably poor, raising some doubts. Moreover, Figure 5(a) demonstrates that with noise, the comparison methods experience a performance drop in CFP-FP during the latter stages of training. Why not use the model from before overfitting occurred as the final model?
3. The test set seems outdated, considering the emergence of many new challenging benchmark datasets in recent years, such as ICCV-MFR (http://iccv21-mfr.com/).
4. It would be beneficial to provide the IDs of the noisy samples from the MS-Celeb-1M dataset to directly illustrate how the proposed method can effectively distinguish between difficult and noisy samples.
5. It is suggested that the paper should analyze the limitations of the proposed method.

**Suitability:**

3

---

### Meta-Review · Area_Chair_6t44 · 2024-07-07

**Recommendation:** Accept (Poster)
**Confidence:** 4

**Metareview:**

Four of the reviewers consistently made the final rating as 3 borderline accept and one accept. There are some concerns raised in the initial round, such as lacking literature comparison, the analysis of easy, hard and noisy samples. The authors rebuttal successfully addressed the concerns from the reviewers. The authors are encouraged to put all the rebuttal improvement materials into the final version of the paper. The overall rating is accept.